# Competition between crystal and fibril formation in molecular mutations of amyloidogenic peptides

Nicholas P. Reynolds [1], Jozef Adamcik[2], Joshua T. Berryman[3], Stephan Handschin[2],
Ali Asghar Hakami Zanjani[3], Wen Li[4], Kun Liu[4], Afang Zhang[4] & Raffaele Mezzenga[2,5]

Amyloidogenic model peptides are invaluable for investigating assembly mechanisms in disease related amyloids and in protein folding. During aggregation, such peptides can undergo bifurcation leading to fibrils or crystals, however the mechanisms of fibril-to-crystal conversion are unclear. We navigate herein the energy landscape of amyloidogenic peptides by studying a homologous series of hexapeptides found in animal, human and disease related proteins. We observe fibril-to-crystal conversion occurring within single aggregates via untwisting of twisted ribbon fibrils possessing saddle-like curvature and cross-sectional aspect ratios approaching unity. Changing sequence, pH or concentration shifts the growth towards larger aspect ratio species assembling into stable helical ribbons possessing mean-curvature. By comparing atomistic calculations of desolvation energies for association of peptides we parameterise a kinetic model, providing a physical explanation of fibril-to-crystal interconversion. These results shed light on the self-assembly of amyloidogenic peptides, suggesting amyloid crystals, not fibrils, represent the ground state of the protein folding energy landscape.

[1] Swinburne University of Technology, ARC Training Centre for Biodevices, Faculty of Science, Engineering and Technology, John Street, Melbourne, VIC 3122, Australia. [2] ETH Zurich, Department of Health Sciences & Technology, Schmelzbergstrasse 9, LFO, E23, 8092 Zürich, Switzerland. [3] University of Luxembourg, Department of Physics and Materials Science, 162a Avenue de la Faïencerie, Luxembourg City L-1511, Luxembourg. [4] Shanghai University, Department of Polymer Materials, Nanchen Street 333, Shanghai 200444, China. [5] ETH Zurich, Department of Materials, Wolfgang-Pauli-Strasse 10, 8093 Zurich, Switzerland. Nicholas P. Reynolds, Jozef Adamcik and Joshua T. Berryman contributed equally to this work. Correspondence and requests for materials should be addressed to R.M. (email: raffaele.mezzenga@hest.ethz.ch)

Amyloid fibrils are implicated in over 20 neurodegenerative diseases, with no curative therapies available to date. A number of amyloids with beneficial functions, the so-called functional amyloids, have also been discovered in biology[1]. Furthermore, amyloids have found applications in technological fields as diverse as tissue engineering, energy, biosensing and water treatment[2–5].

Short amyloidogenic peptide sequences identified from disease-related proteins are highly valuable as reductionist models to study molecular organisation in amyloids[6–11]. Structural diversity can lead to a rich variety of morphologies including filaments[12], nanotubes[13], helical ribbons[6,14], twisted ribbons[14] and crystals[6–9,15],

a phenomenon known as amyloid polymorphism. The availability of crystalline polymorphs is particularly important as it allows, via X-ray crystallography, the precise determination of molecular organisation in amyloid-like species[7,8]. Currently, the mechanisms governing fibril-crystal interconversion are unclear and a number of different hypothesis have been proposed including, differing side chain packing[9], nucleation of fibrils from crystals[7], fragmentation or dissolution of one polymorph to the other[9]. Without a detailed knowledge of the mechanisms leading to crystal formation it is impossible to definitively link the crystal structures with those of more disease-relevant fibrils or protofilaments[9]. A molecular understanding of the amyloid crystallisation mechanism is also of

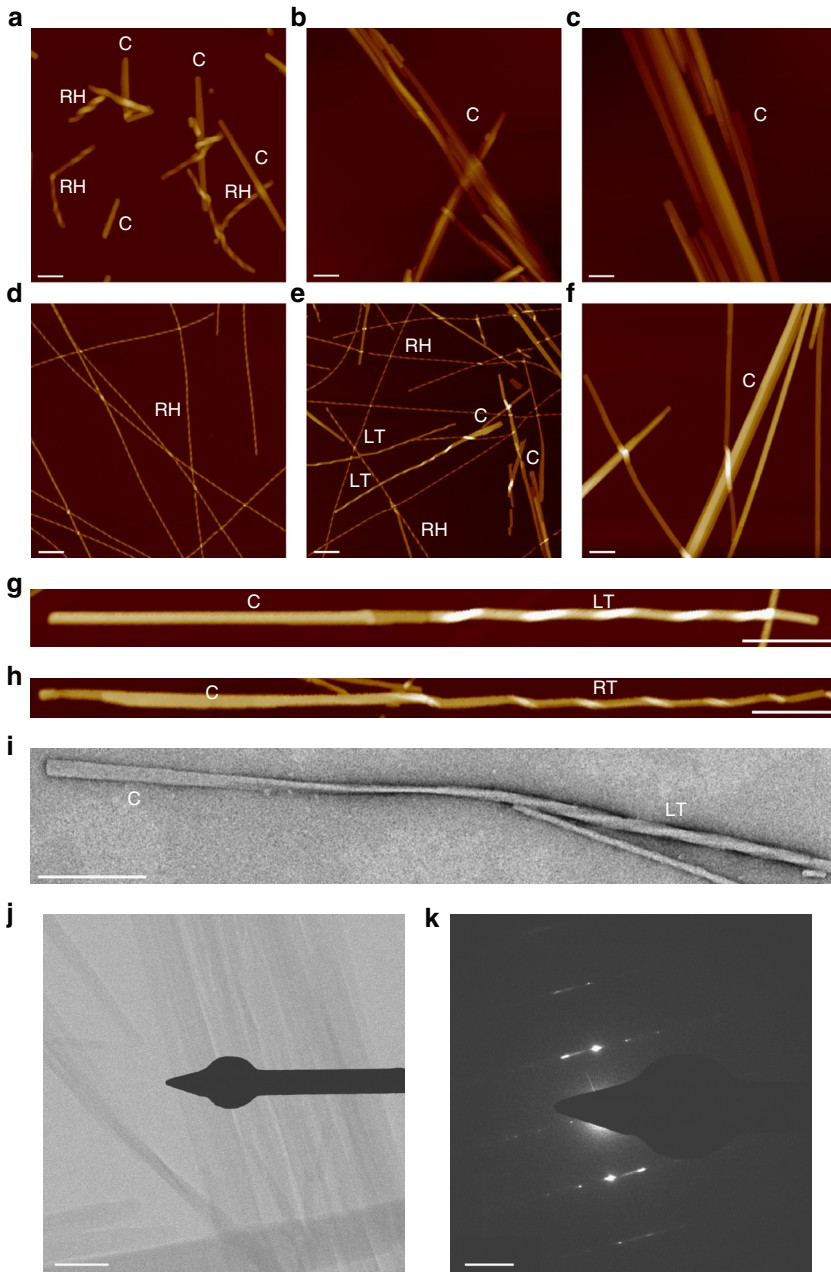

**Fig. 1** AFM and TEM images of self-assembled structures formed by the three hexapeptides. **a** AFM images of ILQINS (Z-scale = 100 nm), **b** IFQINS (Z-scale = 100 nm) and **c** TFQINS (Z-scale = 100 nm) at pH 2 and **d** ILQINS (Z-scale = 30 nm), **e** IFQINS (Z-scale = 30 nm) and **f** TFQINS (Z-scale = 100 nm) at pH 7. AFM images **g**, **h** (Z-scale = 20 nm) and TEM image (negatively stained) **i** of IFQINS at pH 7 showing fibril-crystal conversion (scale bars = 200 nm). All structures formed at 1.5 mM peptide concentration after a 24 h incubation. RH = right-handed helical ribbon, LT = left-handed twisted ribbon, C = Crystal. **j** TEM images of TFQINS aggregates (scale bar = 200 nm) where **k** SAD was performed confirming the crystallinity of the TFQINS structures (scale bar = 20 nm$^{-1}$)

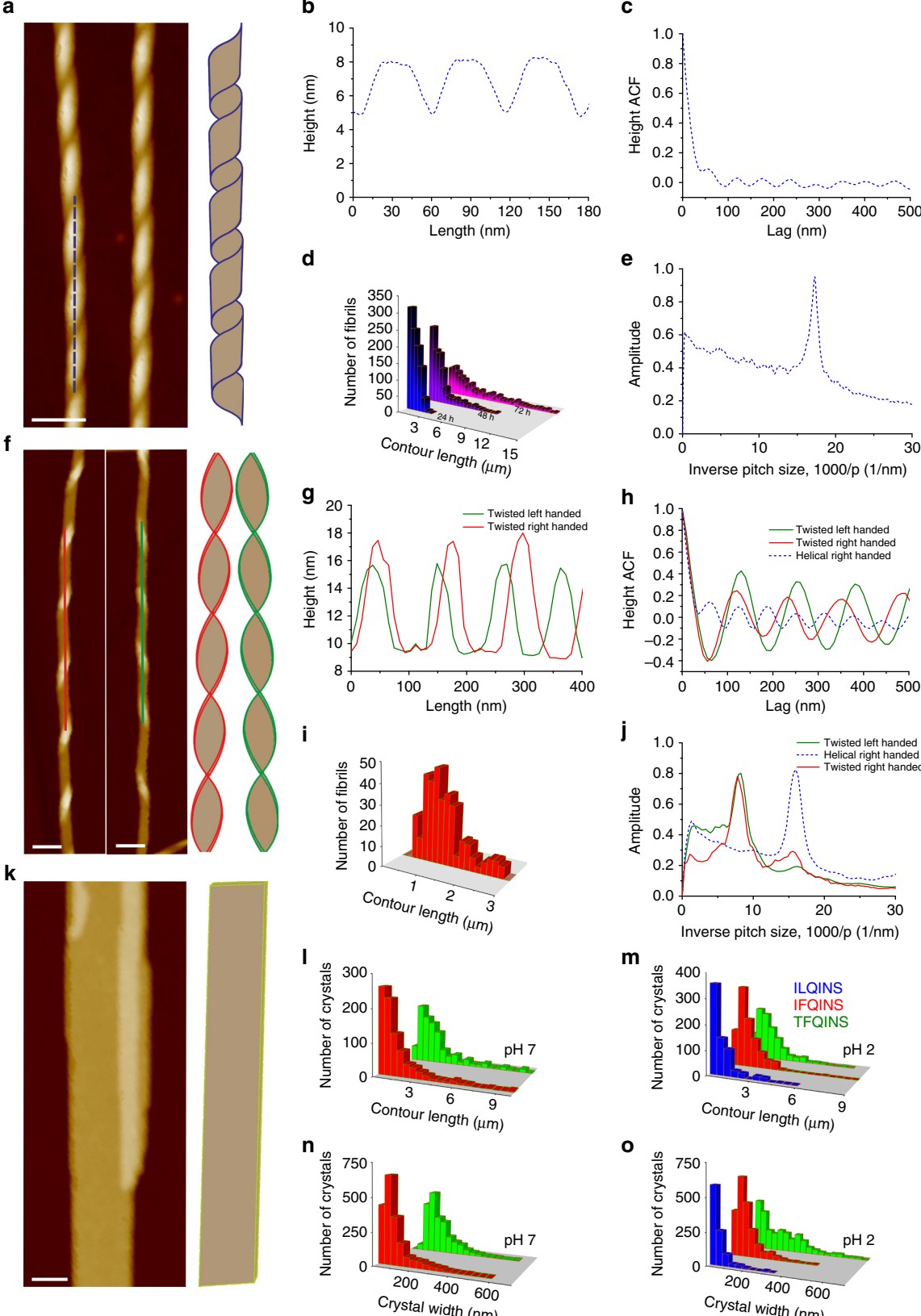

**Fig. 2** Statistical analysis of helical fibrils, twisted fibrils and crystals. **a–e** Helical fibrils from ILQINS, **f–j** twisted fibrils of IFQINS and **k–o** crystals of all three peptides. AFM height images **a**, **f**, **k** (scale bar = 50 nm, Z-scale = 20 nm) with corresponding height profiles **b**, **g**. Contour length distribution of fibrils **d**, **i**; estimation of the pitch calculated from the autocorrelation function (ACF) **c**, **h** and discrete Fourier transform (DFT) **e**, **j**. Statistical analysis of TEM data sets (~1000 crystals) showing the distribution of crystal contour lengths **l**, **m** and widths **n**, **o** at pH 2 and pH 7

fundamental importance in the field of protein folding, allowing us to place the amyloid crystal in the protein folding energy landscape of which the amyloid fibril has been hypothesised to be the ground state[16,17].

Here we uncover the molecular and mesoscopic mechanisms leading to fibril-crystal conversion by using judicious mutations in a homologue series of amyloidogenic hexapeptides, ILQINS, IFQINS and TFQINS, identified, respectively, from hen egg white lysozyme[6], human lysozyme and a mutated form known to cause hereditary systemic amyloidosis[18]. Specifically, we show that the structural characteristics of the fibrils determine if fibril-crystal conversion can occur, with only twisted ribbon fibrils with saddle-like curvature able to convert into crystals, whereas mean-curvature helical ribbons remain in a fibrillar morphology. We also capture snapshots of fibril-crystal conversion within the same individual object, indicating that conversion occurs through mesoscopic rearrangements of the same peptide aggregate. By combining high-resolution atomic force microscopy (AFM) and X-ray diffraction with statistical physical concepts and molecular dynamic simulations, we propose a molecular unit cell of the aggregated species and establish how intrinsic variables such as pH and concentration guide this polymorphic conversion, quantitatively describing this long-standing important issue in amyloid self-assembly.

## Results

**Twisted amyloid fibrils untwist into amyloid crystals**. The morphology of the self-assembled structures generated from our hexapeptides was characterised on both mica and highly ordered pyrolytic graphite (HOPG) substrates (by AFM) (Fig. 1, Supplementary Figs. 2, 3), on carbon coated grids (by transmission electron microscopy, TEM) (Fig. 1, Supplementary Fig. 4) and in a hydrated, substrate-free environment (cryo-TEM, Supplementary Fig. 5). At pH 2 (1.5 mM peptides) the majority of assemblies adopted a flat, amyloid morphology with no identifiable macromolecular chirality, otherwise known as an amyloid crystal[8,19], however a few ILQINS helical ribbons remained (Fig. 1a). At pH 7 the tendency to form crystals was reduced, we observed exclusively right-handed helical ribbons for ILQINS; a mixture of right-handed helical ribbons, left and right-handed twisted ribbons and crystals for IFQINS; and exclusively crystals for TFQINS. Most importantly, for IFQINS (pH 7, 1.5 mM) we frequently observed the twisted ribbons untwisting and converting to a crystalline morphology (Fig. 1g–i, Supplementary Fig. 6). Such a conversion was never observed for any of the right-handed helical ribbons. Identical morphologies and ribbon untwisting were seen on all substrates (HOPG, carbon and mica, Fig. 1, Supplementary Fig. 6) and substrate-free conditions (cryo-TEM, Supplementary Fig. 6), providing compelling evidence that the observed phenomena are not influenced by the substrate used for imaging. The crystalline nature of the untwisted species observed was confirmed by observing distinct diffraction patterns in electron selected area diffraction (SAD) analysis of TFQINS and IFQINS crystals imaged by TEM (Fig. 1j–k, Supplementary Fig. 7e–h), while no diffraction patterns were observed from fibrillar assemblies (Supplementary Fig. 7i, j). Additionally, diffraction contrast (Supplementary Fig. 7a–d) was observed in brightfield cryo-TEM images from both fully untwisted ribbons and the twisted ribbons in the process of crystalline conversion (Supplementary Fig. 7a–c). On the other hand, we never observed any diffraction contrast from helical ribbons (Supplementary Fig. 7d).

**Stable helical ribbons and labile twisted ribbons**. Using our own-developed open source analysis software FiberApp[20], we performed a statistical analysis of the right-handed helical

ribbons, the (predominantly left-handed) twisted ribbons, and the crystals formed from the three hexapeptides (Fig. 2). The analysis showed that the right-handed helical ribbons formed by ILQINS at pH 7 were highly ordered structures with a well-defined pitch of 60 nm. We further confirmed the stability of the helical ribbons by incubating them for up to 72 h (1.5 mM and pH 7), without any observed crystal conversions (Fig. 2d). The twisted ribbons formed by the IFQINS peptides were less ordered with multiple peaks and a broader dominant peak in the plot of amplitude versus inverse pitch size (Fig. 2j) corresponding to a major population with a pitch of 125 nm and a second population varying between 55 and 76 nm. Analysis of our crystals formed at pH 2 and pH 7 showed increased contour length and crystal width (Fig. 2i–o, Supplementary Table 1) following the trend TFQINS > IFQINS > ILQINS matching the tendency for increased crystallisation as seen in Fig. 1.

The results from the microscopy study clearly showed the presence of three polymorphs, the twisted ribbon, the helical ribbon and the crystal. They also provide compelling evidence that the L-F (in ILQINS-IFQINS) and I-T (in IFQINS-TFQINS) mutations progressively increase the tendency to form larger crystals. These crystals are composed of laterally aggregated untwisted ribbons that are a stabilised form of less ordered, unstable twisted ribbon precursors. The helical ribbons on the other hand have a well-ordered structure that remains stable in their fibrillar structure. This can be easily rationalised by considering the different energetic levels of twisted versus helical ribbons in amyloid-forming peptides. It is well-understood that for cross-sectional aspect ratios of the order of unity, twisted ribbons are more energetically stable and store elastic energy by pure torsion, as revealed by their characteristic saddle-like curvature[21–23]. Beyond a critical cross-sectional aspect ratio, twisted ribbons transform into helical ribbons, undergoing a major topological change from saddle-like to mean-like curvature objects, and storing energy by a combination of both torsion and bending (with an additional torsion-bending coupling term in the energy functional, responsible for chiral symmetry breaking)[21–23]. This captures very well the energetic essence of the observed transition of twisted ribbons to crystals (via an untwisting event), which for helical ribbons would require the much more expensive untwisting and unbending, and is therefore not observed experimentally.

**Molecular organisation is conserved in all structures**. To gain insight into the molecular organisation of the three peptides at pH 2 and pH 7, we performed Wide Angle X-Ray Scattering (WAXS) measurements (1.5 mM, 24 h) (Fig. 3a–c, Supplementary Fig. 8). We observed that WAXS spectra for all three peptides possessed five main Bragg reflections at 0.32, 0.38, 0.49, 0.63 and 1.32 Å$^{-1}$ (corresponding to d-spacings of 19.6, 16.5, 12.8, 9.9 and 4.8 Å). Increased intensities of the reflections in the WAXS spectra for TFQINS compared to the other two peptides were observed at both 24 h (Supplementary Fig. 8, Supplementary Table 2) and 60 s (Supplementary Fig. 9), matching the rapid assembly and increased crystal sizes seen for TFQINS (Fig. 2, Supplementary Fig. 10, Supplementary Table 1).

The reflections at 1.32 Å$^{-1}$ (**001**) and 0.63 Å$^{-1}$ (**020**) were promptly assigned to the intramolecular β-sheet and inter-sheet spacing, respectively[24]. In previous analyses of ILQINS WAXS data, it was not possible to fit both smaller peaks in the centre of the area $0.3 < q < 0.65$[6]. To fit these peaks here, it was hypothesised that they correspond to **110** reflections from multiple configurations having different values of the unit cell angle γ. The **110** peak is expected to respond more strongly to variation in γ than the **n00** or **0n0** peaks, which each depend

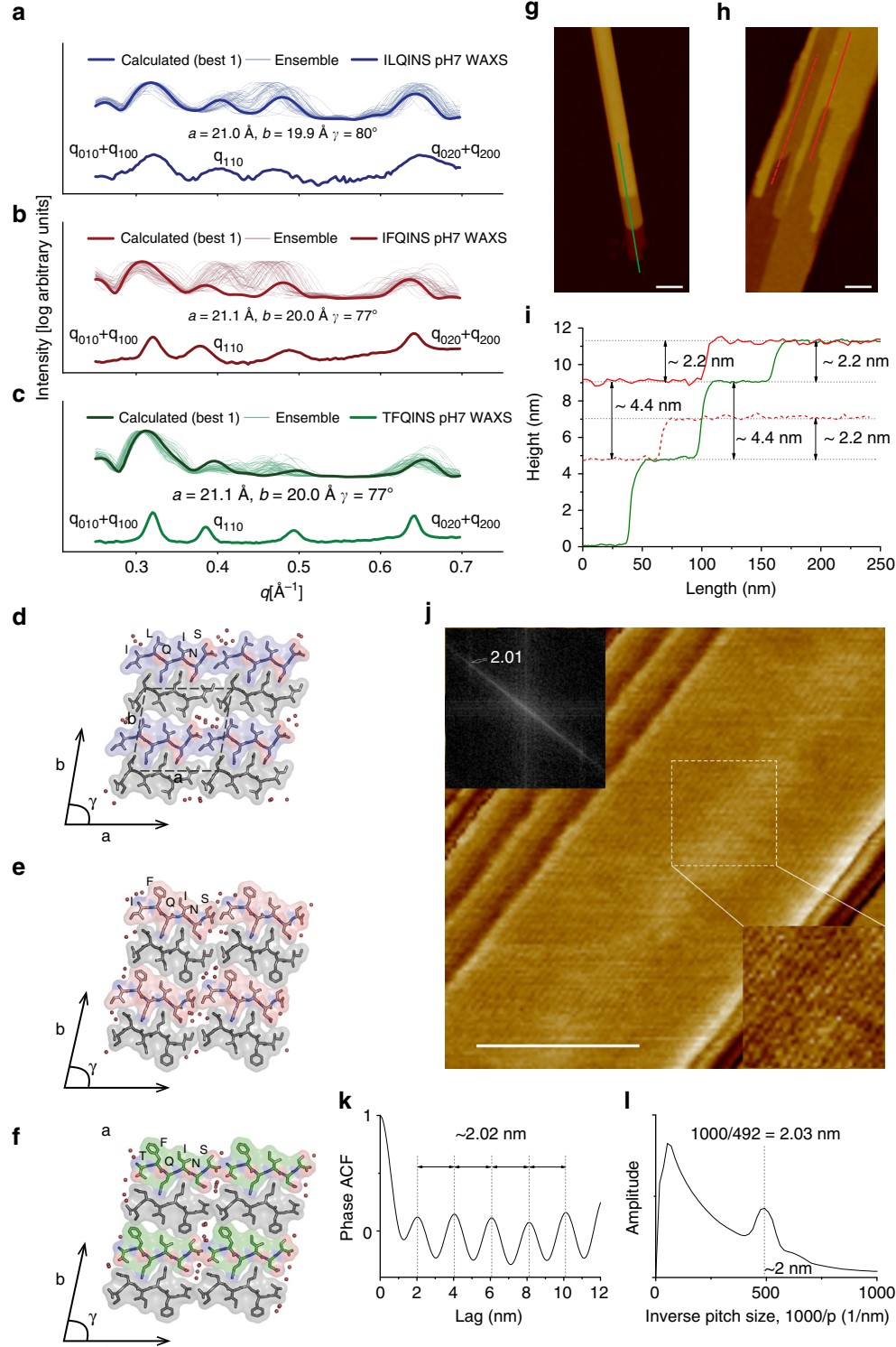

**Fig. 3** WAXS, AFM and MD simulations shed light on the molecular organisation in the amyloid unit cell. **a–c** Shows calculated scattering curves based on ensembles of nanocrystals. The single curves giving the closest match to the experimental data for each peptide are highlighted, and labelled with their unit cell parameters. The central region of the spectra was highly variable with respect to γ in comparison to the two large peaks at each end. **d–f** show the nanocrystal structures associated with each closest-matching trace, with semi-ordered dynamic water molecules shown as orange spheres. AFM images of **g** TFQINS (Z-scale = 30 nm) and **h** IFQINS (Z-scale = 30 nm) with **i** the corresponding height profiles showing that the height of one single layer is 2.2 nm. **j** AFM phase images of TFQINS showing lateral periodicity of ca. 2 nm (FFT inset in left corner). Scale bars are 50 nm. **k, l** show the evaluation of lateral periodicity by **k** averaged ACF of phase image and **l** DFT analysis of phase images

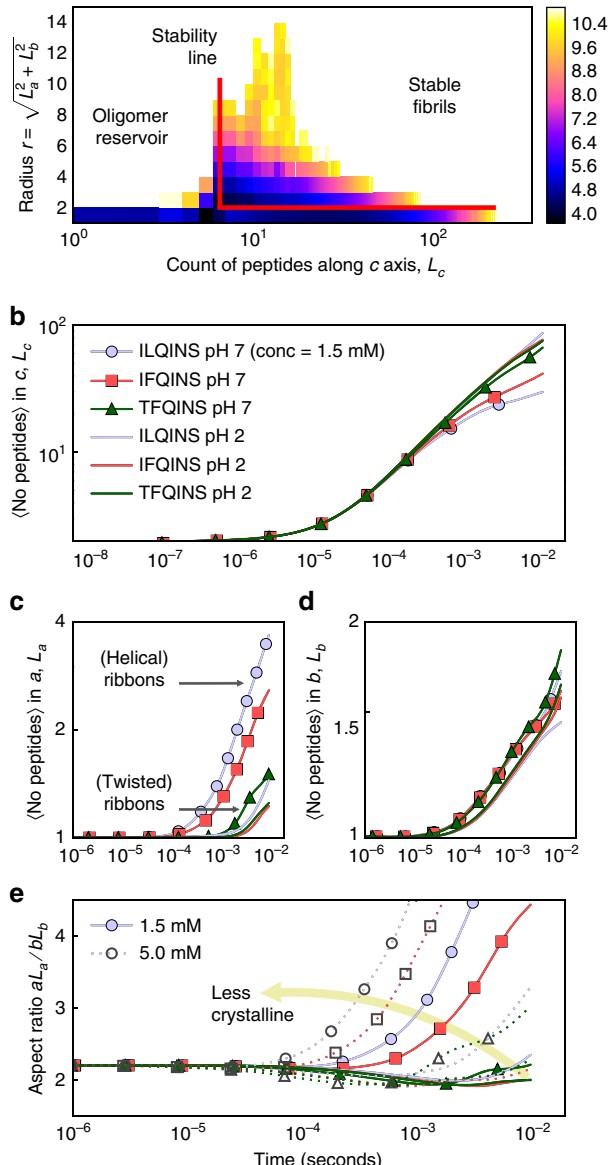

**Fig. 4** Evolution of oligomer formation along axial (*c*) and lateral (*a*, *b*) assembly axes. **a** Two-dimensional histogram of length-radius distribution for TFQINS peptide aggregates at pH 2 after 0.5 ms. Lateral aggregates are not stable below a barrier length of $L_c = 6$ peptides. **b** Plot of axial aggregation against time showing that ILQINS and IFQINS slow their axial aggregation. **c** Plot of lateral aggregation in *a* versus assembly time, and **d** plot of lateral aggregation in *b* versus assembly time. Aggregation in *a* takes off sharply at about 1 ms while lateral aggregation in *b* starts at 0.1 ms but proceeds steadily for all systems. **e** Cross-sectional aspect ratio of the growing oligomers against time: the spread of aspect ratios corresponds very closely to the spread of crystal formation in the experimental samples

primarily either on the *a* or *b* only. It was observed in simulations that $\gamma$ is indeed a soft degree of freedom for the nanocrystalline model systems (Supplementary Movies 2, 3).

For each sequence, an ensemble of crystal structures was generated for the range $74° \leq \gamma \leq 91°$ (Supplementary Methods). Figure 3a–c shows that at least one scattering profile with a good match to the experimental data were always among those generated. While the central peak associated to **110** did prove to depend very strongly on $\gamma$, it showed a tendency not only to move but also to split into two, indicating that other potential

contributions depending on $\gamma$ might also be present and generating complex interference phenomena. Variation between structures was associated mostly with shear at the hydrophobic-hydrophobic sidechain interface (Fig. 3d, e, Supplementary Movies 2, 3) and the replacement of bulky sidechains with an increase in the amount of (semi-ordered, dynamic) water (Fig. 3d–f). The overall contact topology was largely unaffected.

The dimensions of simulated unit cells (in *a*, *b*, *c*) giving the closest match to the experimental scattering were found to be of monoclinic symmetry with: 21.0, 19.9, and 4.9 Å, 21.1, 20.0 and 4.9 Å, and 21.1, 20.0 and 4.9 Å for ILQINS, IFQINS and TFQINS, respectively. We further assessed the simulated unit cell by high-resolution AFM. The lateral periodicity of TFQINS protofila-ments (*b*) was estimated around 20 Å (Fig. 3j–l), and the height of single crystal layer (*a*) was estimated around 22 Å (Fig. 3g–i and Supplementary Fig. 11) for both TFQINS and IFQINS. These values are remarkably close to the dimensions of the unit cell resolved by WAXS and molecular dynamic simulations (≤1 Å between experiment and simulation).

**Parameterisation of a kinetic model of assembly**. Based on our atomistic models, we calculated the free energy change for lateral aggregation of small oligomers of the three peptides via molecular dynamic simulations. The free energy changes ($-\Delta G$) were decreasingly favourable in the order IL > IF > TF and the order pH 7 > pH 2 (Supplementary Table 3). This is in agreement with physical expectation as threonine is not expected to be a strongly amyloidogenic residue due to its lack of both hydrophobicity and steric bulk. We observed individually small free energy gains from lateral association with respect to the number of peptides buried at the interface, leading us to conclude that individual peptides cannot form strong laterally associated dimers. However, pairs of $\beta$-structured oligomers above a length of ~6–10 strands of any peptide in the fibril axis direction (*c*), have a large cumulative free energy gain for lateral association due to the large face area, so will associate irreversibly along the *a* or *b* axes via terminus-terminus or sidechain-sidechain interactions (Supplementary Fig. 12). This provides us with an expected kinetic scheme, which bears a resemblance to the common scheme of amyloid growth kinetics involving nuclei formation (the lag phase) followed by rapid fibril extension. Specifically our scheme involves initial association into short (≥6–10 peptides) $\beta$-structured oligomers, which will be diffusion-limited and largely unaffected by muta-tions; followed by further axial extension (*c*) of the formed oligomers combined with lateral assembly in *a* and *b*. This lateral assembly of oligomers will be delayed until larger length in *c* by the I-T mutation, by low pH, or by low concentration.

With the energy for lateral association as input, we can gain a molecular understanding of the propensity of the peptides to form twisted versus helical ribbons, by monitoring the evolution of the cross-sectional aspect ratio ($aL_a/bL_b$) from its earliest stages and thus, resolving the diverse tendencies to ultimately form crystals. Figure 4 shows the results of a Doob-Gillespie simulation[25] of 1 million peptides per system at different concentrations (1.5 and 5 mM).

In Fig. 4a we show a snapshot of the distribution of aggregates of length $cL_c$ and radius *r*. Once the oligomers pass a certain length ($L_c > 6$, $cL_c > 28.8$ Å) we see increased probability of oligomers being formed with larger radii. This is due to the increased free energy of lateral association with increasing surface area of the longer aggregates. Figure 4b shows that during the initial stages of aggregation, stronger inter-peptide attraction due to pH or sequence results in a slowing (ILQINS > IFQINS > TFQINS, ILQINS being the slowest) of axial aggregation (the *c* axis). This corresponds with an increase in earlier lateral

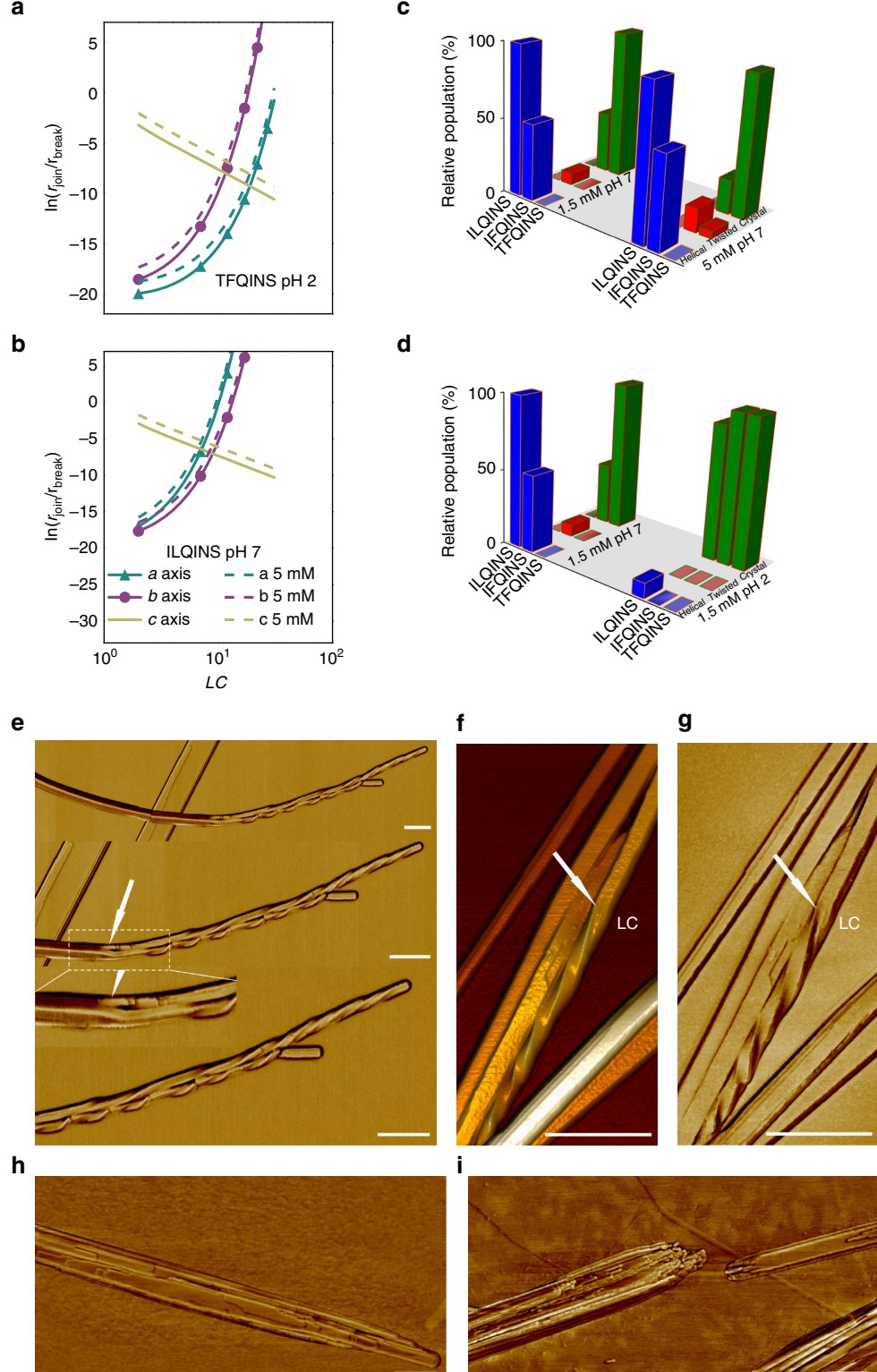

**Fig. 5** Theoretical and experimental proof that increased peptide concentration reduces crystal formation. Analytical rate calculations for **a** TFQINS at pH 2 and **b** ILQINS at pH 7 showing the ratio of forward and backward rates of assembly against increasing length in the (initially dominant) c direction. **c**, **d** Compare the composition of polymorphs observed experimentally (by AFM) for the three peptides, **c** shows the effect of varying concentration at pH 7, and **d** shows the effect of varying pH at 1.5 mM. **e** AFM phase images of TFQINS deposited on mica self-assembled at pH 7 (5 mM) for 24 h showing splitting of crystal into two twisted fibrils. AFM 3D image **f** and AFM phase images **g** of TFQINS deposited on mica self-assembled at pH 7 (5 mM) for 24 h showing fibril-crystal transformation. AFM phase image of TFQINS deposited on **h** mica and **i** HOPG self-assembled at pH 7 (5 mM) for 24 h showing lateral aggregation (scale bar = 200 nm)

aggregation (terminus–terminus) in the $a$ axis (Fig. 4c), but has little effect on aggregation in the ($\beta$-sheet) $b$ axis (Fig. 4d), leading to the formation of anisotropic oligomers with high cross-sectional aspect ratio ($aL_a/bL_b \gg 1$). The simulations suggest that this tendency to form anisotropic oligomers is increased not only by hydrophobic sequence and by more neutral pH but also by increasing concentration as shown in the plot of aspect ratio versus time (Fig. 4e).

For further insight into the kinetics, we can make an approximate treatment of the early stages of aggregation by assuming a homogenous solution of $N/L_c$ linear oligomers each of length $cL_c$. Figure 5a, b shows the log ratios of forward and backward rates of assembly in 3 dimensions $a$, $b$ and $c$ (equivalent to the free energy difference, in an equilibrium system) for two systems having opposite characteristics, namely TFQINS at pH 2 (Fig. 5a) and ILQINS at pH 7 (Fig. 5b). In all cases the rate of assembly in $c$ is the largest, but for conditions where crystals dominate (Fig. 5a) the rate of assembly in $b$ is larger than the rate of assembly in $a$. Conversely in conditions that promote fibril formation (Fig. 5b) the rate of assembly in $a$ becomes larger than in $b$: the former case leads to isotropic cross-sections, whereas the latter yields anisotropic ones. This is because the addition of a molecule along $b$ produces an increase in spacing of the order of a $\beta$–sheet ($\approx 1$ nm), whereas the addition of a molecule along $a$ increases spacing by the contour length of the molecule ($\approx 2$ nm). Figures 1, 2 show that at much longer timescales under the conditions simulated in Fig. 5b helical ribbons made up of anisotropic oligomers[21–23] are the dominant species present. If we assume the amyloid systems have a parallel-sheet symmetry of assembly, predicted to be the most common $\beta$-sheet organisation[26] then two-dimensional structures where assembly in $c$ and $a$ dominates will have mismatched hydrophobicities between faces, due to different sidechain properties. This will lead to the generation of a (mean) bending curvature in the growing fibrils that is amplified as the fibril grows, resulting in mature fibrils with a helical morphology that cannot easily untwist and unbend to aggregate further into a crystal.

The effect of increasing concentration is also shown in Fig. 5a, b using dashed lines to indicate an increase from 1.5 to 5 mM peptide solution. Initial aggregation along the $c$ axis is enhanced in a constant ratio, because the collision cross-section remains constant with $L_c$, however the secondary aggregation axes are not enhanced to the same extent, as the evolution of the rate ratio has more complex dependence on the aggregate size and shape.

The increased formation of anisotropic oligomers, at early time points ($<0.1$ s), due to promotion of lateral aggregation in response to pH (pH 7 > pH 2) and chemical sequence (ILQINS > IFQINS > TFQINS) predicted by the simulations is consistent with the experimental results collected at much longer timescales (hours/days). We see the same trends occurring when looking at the extent of helical ribbon formation versus twisted ribbon (leading to crystal development). The vital pivot in the kinetic route is the higher cross-sectional aspect ratio of helical ribbons (more anisotropic) compared to twisted ribbons, which typically possess cross-sectional aspect ratios approaching unity[21–23].

With this in mind, it then becomes possible to test experimentally the last parameter favouring the formation of anisotropic oligomers (and thus stable helical fibrils), which is the increasing concentration, as suggested by the simulations reported in Figs. 4e, 5a, b, and previous observations on assemblies of GNNQQNY[27]. In Fig. 5c, d we monitored solutions of the peptides at 1.5 and 5 mM at pH 7 and pH 2. Indeed, both at high pH and high concentrations we observed a reduced population of TFQINS and IFQINS crystals and a corresponding increase in twisted and helical ribbon structures (Fig. 5c, d and Supplementary Figs. 13, 14). Furthermore, at these increased

concentrations, we now even observe a few events of TFQINS twisted ribbon to crystal interconversion (Fig. 5e, g). At lower concentrations (Fig. 1), we saw no evidence of fibril-crystal conversion for TFQINS, as all interconversions had already taken place. These results validate our predictions that increased aggregation in the axial and in the sidechain-sidechain lateral directions (leading to crystal formation) is promoted when growth in the terminus-terminus lateral direction is thermodynamically penalised.

The identical peak positions in the WAXS spectra for all species formed and the accurate reproduction of the peak positions in the simulated WAXS spectra of the molecular models of all three peptides suggest that helical or twisted ribbons and crystals all have very similar molecular structures. As the amyloid crystals are significantly wider (Fig. 2) than the helical or twisted ribbons they must be composed of many laterally aggregated (un)twisted ribbons. Indeed, we see evidence of this in the AFM and TEM images, where we can clearly resolve both laterally aggregated protofilaments (Fig. 3j, Supplementary Figs. 11, 14) and on larger length scales laterally aggregated (stacked) crystals (Figs. 3g, h, 5e–i, and Supplementary Fig. 15).

To summarise we have uncovered a mechanism of fibril-crystal interconversion in amyloidogenic peptide systems and have identified the physical driving forces ruling this structural transition. In the homologue peptide series studied, crystal formation occurs via the mesoscopic untwisting and lateral aggregation of twisted ribbon precursors, without noticeable changes in X-ray diffraction patterns. This provides structural evidence that the amyloid crystal structures elucidated before[7–9] may have relatable molecular structures to the more disease-relevant fibrillar polymorphs. More importantly, these same results may bear a general fundamental significance in protein folding. Previously amyloid fibrils were postulated to be the ground state in the protein folding energy landscape[16,17], however our results suggest that they can further reduce their energy state by releasing torsional energy thus converting to amyloid crystals. This implies that amyloid crystals and not amyloid fibrils may occupy the absolute minimum in the protein folding energy landscape. This is of course conditional on the existence of such a crystalline minimum in the protein folding energy landscape: while the results presented here clearly demonstrate its existence for short amyloidogenic peptides, amyloid crystalline morphologies are known to occur also in longer peptide sequences up to 11-mers[28] and even 26-mers fragments from tau protein (Supplementary Fig. 16). These sizes exceed those of short native proteins (Trp-Cage is 18–20 residues[29]) and hormones (Somatostatin is 14–28 residues[30]) capable of folding correctly. For longer proteins, however, amyloid crystals may become progressively less accessible, due to entropic restrictions.

The mechanism of fibril-crystal conversion discussed here and its energetic description, unify, expand and advance previous theoretical and experimental studies of fibril-to-crystal conversion. Specifically, the transition from twisted fibrils to crystals via a progressive untwisting, is consistent with the proposed Landau energetic description connecting the two states as a sole function of the twisting angle[19], although the lack of torsion-bending coupling terms in such a Landau expansion means that the helical fibril intermediates observed experimentally here cannot be accounted for. From the experimental standpoint, the progression of the conversion from fibrils to more energetically stable crystals observed for ILQINS, IFQINS and TFQINS is consistent with the systematic switch of fibrils to crystals observed in GNNQQNY[9], with the additional key finding that the two energetic states can be sampled by simply releasing torsional elastic energy while maintaining identical molecular packing, suggesting the energy barrier separating the two states to be very small.

## Methods

**Transmission electron and atomic force microscopy.** Both room temperature (RT) and cryogenic transmission electron microscopy (cryo-TEM) were performed using a Tecnai 12 transmission electron microscope; diffraction patterns were imaged using a Tecnai F20 (FEI, Eindhoven, The Netherlands). Images were recorded on a Megaview III CCD camera (RT-TEM), a FEI Eagle 4k × 4k CCD camera (cryo-TEM), or a Gatan US4000 4kx4k CCD camera (diffraction patterns). For RT-TEM, assemblies were negative stained with potassium phosphate (2%, pH 7.2, 10 s). Samples for diffraction were prepared at room temp and cooled down for measuring (−180 °C). Cryogenic samples were prepared in liquid ethane using a Gatan 626 cryoholder (Gatan, Pleasanton, CA, USA). Atomic Force Microscopy (AFM) experiments were performed on a Nanoscope VIII Multimode Scanning Force Microscope (Bruker) operated in tapping mode in air. Image processing (first order flattening) was performed in the NanoscopeAnalysis 8.15 software, and statistical analysis was performed using the open source software FiberApp[20]. More details are provided in Supplementary Methods.

**Peptide synthesis.** The three hexapeptides were synthesised by standard solid phase peptide synthesis according to synthetic procedures developed previously[6]. More details are provided in Supplementary Methods.

**Wide angle X-ray scattering.** Wide Angle X-ray Scattering (WAXS) was performed on the SAXS/WAXS beamline at the Australian synchrotron. Spectra were recorded at RT using a beam of wavelength $\lambda = 1.03320$ Å (12.0 KeV) with dimensions 300 μm × 200 μm and a typical flux of $1.2 \times 10^{13}$ photons per second. 2D diffraction images were recorded on a Pilatus 1 M detector. Experiments were performed at $q$ ranges between 0.03 and 1.5 Å$^{-1}$. Spectra were recorded under flow (0.15 ml min$^{-1}$) in order to prevent X-ray damage from the beam. Multiples of 15 spectra were recorded for each time point (exposure time = 1 s) and the averaged spectra are shown after background subtraction against MQ water or MQ water at pH 2 in the same capillary.

**Atomistic calculation of desolvation energies.** The AMBER simulations were run using the AMBERFF14-SB all-atom forcefield[31]. When the C-terminus was neutralised to mimic conditions of low pH, the same partial charges for the protonated C-terminal serine were used as in Lara et al.[6]. Structures were based on the ILQINS pH 2 structure previously validated against scattering[6], with point-mutations carried out using pymol without disruption to the unit cell geometry. Predicted WAXS spectra were generated from the MD trajectories using Crysol[32]. See Supplementary Methods for further information.

Binding free energies for interfaces in the $a$ (peptide axis), $b$ (β-sheet axis) and $c$ (fibril axis) of the simulated microcrystal structures were found by calculating the free energy of blocks of peptides separately and joined together, eg: $\Delta G_a^o = G_{266}^o - (G_{166}^o + G_{166}^o)$, where subscript triplets are the dimensions of a peptide block or sub-block. 1000 times $3 \times 6 \times 6$-peptide parallelpiped blocks were sampled from large molecular dynamics simulations of each system. Block free energies were estimated as averages over the 1000 snap sets using AMBER with the solvation part of the free energy estimated using a generalised Born method[33]. The Supplementary Methods contains further information.

**Doob-Gillespie.** The input to a Doob-Gillespie calculation[25] is a system of forward and backward rate constants for association between the different species present in solution. Individual reactions are then carried out on a stochastic basis. To reduce the number of distinct species which need to be tracked, we consider only processes for assembly or splitting of rigid assemblies of $\beta$-structured peptides, neglecting conformational changes.

The rate for association of two oligomers $i$, $j$ having matching-sized planes in $a$ can be defined *via* the diffusive collision rate for anisotropic bodies with surface $bL_bcL_c$ in a volume $V$, subject to an exponential barrier of $3k_BT$ representing the entropic cost to remove translational and rotational degrees of freedom by joining two rigid bodies. From this, we are able to define a manageable system of Arrhenius-type rate equations, suitable for Monte Carlo sampling via the Doob-Gillespie method. Here rate equations are presented in the form directly useful for simulation, such that the units are s$^{-1}$:

$$r_{aa} = \frac{2N_iN_j}{V} e^{-3} (D_i + D_j) \sqrt{2bL_bcL_c}. \tag{1}$$

The diffusion constants $D_i$, $D_j$ for the two bodies are defined as for rod-like objects[34].

The rate for a cuboidal oligomer of $L_aL_bL_c$ peptides to split along some plane perpendicular to the $a$ crystal axis is defined as:

$$r_a = \frac{1}{\tau_0} N(L_a - 1) e^{\frac{L_bL_c\Delta G_a^o}{k_BT}}. \tag{2}$$

Here $\tau_0$ is a constant which sets the timescale (taken as the single-peptide diffusion time). The term in the exponential rapidly becomes large and negative as the area of the interface to be cleaved grows. Rates for splitting in $b$ or $c$ are available by permuting the indices.

Although simulation of non-cuboidal aggregates was avoided due to the large number of different species which would then need to be tracked, we make an approximation to this by allowing non-matching faces to join providing that the non-cuboidal aggregate formed then immediately splits to form cuboidal products: an area-dependent cost $\Delta$ is added to the exponential barrier to represent the free energy cost of splitting an intermediate non-cuboidal aggregate to then generate two or three cuboidal aggregates as the output of the reaction.

**Data availability.** All data are available from the authors upon reasonable request.

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

## Acknowledgements

N.P.R. would like to thank the Australian Research Council (ARC) Training Centre for Biodevices at Swinburne University of Technology (IC140100023), A.Z would like to thank the National Natural Science Foundation of China (21574078) for financial support, we also thank Dr Nigel Kirby and Dr Adrian Hawley for assistance at the SAXS/WAXS beamline at the Australian Synchrotron. The synchrotron-based research in this work was in part supported by the Science and Industry Endowment Fund (SIEF) Special Research Program-Synchrotron Science. HPC facilities of The University of Luxembourg were used. A.A.H.Z was supported by the *Fonds Nationale de la Recherche* of Luxembourg (C14/MS/8329720). Prof. Tyler Luchko supplied valuable insight and calculations to shape the hypotheses behind this work. Insightful discussions from Dr Stefan Auer are also acknowledged, as well as support by the Center for Optical and Electron microscopy of ETH Zürich (ScopeM) (Dr Fabian Gramm).

## Author contributions

N.P.R. carried out X-ray diffraction, TEM and cryo-TEM, analysed the data and wrote the manuscript. J.A. carried out the AFM analysis. J.T.B. ran simulations, analysed the data and wrote the manuscript. S.H. carried out the SAD experiments. A.A.H.Z. ran simulations. A.Z. and W.L. designed the solid phase peptide synthesis procedure; K.L. synthesised, purified and characterised the peptides. R.M. designed the study, analysed the data and wrote the manuscript. All authors commented on the manuscript.

## Additional information

**Competing interests:** The authors declare no competing financial interests.

**Change history:** A correction to this article has been published and is linked from the HTML version of this paper.

