## [Peer Review File · Nature Communications]

Reviewer #1 (Remarks to the Author):

This work by Reynolds et al. is a thoroughly revised version of a manuscript that I have previously reviewed for Nature Nanotechnology. I have carefully read the rebuttal letter and the revised manuscript. The manuscript has been considerably strengthened by the additional experiments that were performed as well as the more quantitative analysis of the WAXS data that are now compared with simulations. Also, all the points that I had raised previously are addressed satisfactorily.

I would like to congratulate the authors for defining a new benchmark for the depth and quality of their study of short peptide self-assembly and for the combined use of cutting edge experimental and theoretical tools. I recommend publication of the manuscript in the current form.

Reviewer #2 (Remarks to the Author):

In this revised manuscript, the authors have clarified important comments previously raised. They have also added stunning new data to further support their conclusions. As result, I think this manuscript is much improved. Following are few comments on the authors' reply to my previous comments.

1)

"This is an interesting point brought forward by the referee that has inspired us to undertake new work, which we believe, leads to a substantial improvement of the manuscript. To demonstrate that untwisted ribbons are amyloid crystals we now show cryoTEM images (new figure S7) from which electron diffraction occurs in proximity of the untwisted regions but not where the fibrils are twisted.

Furthermore, we have conducted also new, systematic experiments using Selected Area Diffraction (SAD) on flat ribbons (see figure 1 and S7 for representative examples): the flat ribbons lead in all cases to a single diffraction pattern, which as the referee will certainly appreciate, rules out any possible ambiguity concerning the crystallinity of these structures. This is further mirrored by a deeper analysis of WAXS spectra and high-resolution AFM images. Please refer to new figure 3 for a detailed characterization of these crystals. The conformation of crystallinity as proved by SAD observed electron diffraction is discussed on pg4, line 13."

The authors added new SAD images that convincingly show crystal-like order in the proximity of the untwisted regions. They also write that they do not see any diffraction in the areas of the twisted fibrils, but I cannot find these images in the manuscript. The authors should add SAD images in the proximity to the twisted fibrils for comparison. Is it possible to calculate key molecular distances/lattice parameters using the diffraction patterns?

2)

"We are actually in full agreement with the referee and we believe that this point arises from a misunderstanding, as we may have not sufficiently well explained this point in the original manuscript.

Molecular simulations give insight on the trajectories followed by the initial assembly of peptides to build up supramolecular species of given cross-sectional ratio (helical vs twisted ribbons). These simulations shed light on the temporal evolution of growth in the a, b and c axes, and on how the cross-section builds up until the elongation regime dominates the process (see figure 4a) up to

around 10⁻² seconds. Thus, molecular dynamic simulations ONLY give the rationale to establish how the cross-sectional aspect ratio drives the fibrils into either twisted or helical ribbons. These MD simulations allow us to understand how pH, concentration and molecular mutations drive the assembly into either twisted ribbons or helical ribbons and simulations up to 10⁻² seconds are sufficiently long enough to establish this pathway.

The untwisting of twisted ribbons into crystals and the stability of helical ribbons are observed ONLY experimentally at much longer times, as the time scales involved with mesoscopic (and not molecular) rearrangements are much longer. Therefore simulations and microscopy experiments are fully complementary as they probe different time and length scales (MD: molecular length scale and sub-seconds time scales – AFM: mesoscopic length scale and minutes to hours/days time scales, see pg 17, line 1 for a discussion). Once crystals are formed from untwisted ribbons, lateral aggregation of crystallites may proceed without any significant energy barrier and we fully agree with the referee that this can undergo continuous lateral association at any time. One example is the aggregation captured in figure 5e. More examples have been added in the supplementary information (figure S15, discussed on pg 17, line 22). Conversely, lateral aggregation on twisted species, is much more expensive from an energetic point of view, since this implies additional torsional/bending energy.

We therefore use MD simulations to unravel the build up of the aspect ratio in the early aggregates and once the aspect ratio trajectory is identified, we rely on AFM and TEM analysis to study the mesoscopic evolution. As the referee will note, we have, by means of the new AFM analysis, a resolution down to the single constitutive protofilament (figure 3j, S11, and S14). Thus, while the time scales between experiment and simulation cannot be matched, the length scales are now covered from the early molecular addition of the hexapeptides to the mesoscopic evolution of the fibrillar and crystalline species involved, validating the structural evolution followed by the individual aggregates across all the length scales involved in the process.”

The authors are probably correct in this specific case that there is likely a link between the observations made at the different timescales accessible to simulations and experiments, respectively. However, the authors’ evidence is nevertheless indirect, i.e. using information on points A and B to infer a pathway between A and B. Therefore, I suggest the authors to be much more careful in discussing the assembly mechanisms in the manuscript. For example on page 17: “The increased formation of anisotropic oligomers, at early time points (< 0.1 s), due to promotion of lateral aggregation in response to pH (pH 7 > pH 2) and chemical sequence (ILQINS>IFQINS>TFQINS) predicted by the simulations correlates extremely well with the experimental results collected at much longer timescales (hours/days).” Here “correlates extremely well with” should be changed to “is consistent with”.

3)

“Again, we believe that this may be a misunderstanding. We indeed fully agree as the referee is inferring that the free energy barrier separating the two states (fibrils vs crystals) must be small, otherwise the transition would not be observed experimentally (although we have no elements to quantify how much the crystal minima is below the fibril one). We also fully agree that the exact depth of the free energy well of the dominant population must be dependent on the sequence and the conditions employed, and indeed the data presented in the manuscript precisely shows this. We do not understand, however, how the statement formulated above may be in contradiction with these findings: if we are able to capture the spontaneous untwist of fibrils into untwisted crystals without unfolding/refolding of the peptide, the energy of the final state (crystals) must be necessary lower than the initial state (fibril), since the only difference is the released torsional energy associated with the twist (which is positive by continuum mechanics). Therefore the lower minimum of the crystal compared to fibrils is not to be questioned and together with the previously supposed absolute minimum of amyloid fibrils in the protein folding energy landscape, this immediately infers that a new absolute minimum has been found. This is in agreement with Referee 1 and 2 assessment and with earlier findings by Marshall et al. on GNNQQNY. The referee may also want to revisit figure 1 in Hartl et al. Nature Structure and Molecular Biology 2009 in

terms of the favourable energetic transition fibrils → crystals.

We do agree with the referee that the original phrasing may benefit from clarification. To avoid ambiguity, we account for the remark of the referee and rephrase the text accordingly. We have added a conclusive paragraph in the manuscript (pg 19, line 7), stating: "From the experimental standpoint, the progression of the conversion from fibrils to more energetically stable crystals observed for ILQINS, IFQINS and TFQINS is consistent with the systematic switch of fibrils to crystals observed in GNNQQNY3, with the additional key finding that the two energetic states can be sampled by simply releasing torsional elastic energy but maintaining identical molecular packing, suggesting the energy barrier separating the two states to be very small."

Here, the authors' data show that the crystal-like structures are more favourable energetically than twisted filaments, but the difference in energy may be small. Thus, given all the conditions in this discussion that the authors added in their clarification, I think "introducing a new major paradigm" wording is far too exaggerated and should be removed. The authors should also tune down their wording and be more careful on discussing that the crystal-like structures may be energy minima etc.

Reviewer #3 (Remarks to the Author):

The manuscript by Reynolds et al describes the switch that can take place between fibrils and crystals. The paper is well written and I very much appreciate the careful consideration the authors have taken with the reviewer comments. The new section that analyses the WAX data is interesting and informative. I think it could have been even further improved by the added information afforded by fibre diffraction and this might have yielded insights into the differences between the structures. The figure 3a shows comparisons between experimental and calculated data, which shows good fit, but not perfect. However, they are further supported by measurements from AFM. I wonder if the authors have considered the contribution of hydrogen bonding of the Q residues (which has been described before for polyQ peptides). I am not sure where the 4.9 Å comes from? The hydrogen bonding distance for cross-beta structures is 4.76Å, but is often quoted as 4.7 or 4.8.

Finally, I appreciated all the changes to the paper, but I do not think that new Figure S16 is needed here. The authors claim (in their response letter) that this is the first crystal of 26 residues. However, whilst other crystal structures have not been solved (using Xray crystallography), several other amyloidogenic proteins/peptides form crystalline structures. PolyQ50 forms crystals amongst others. Figure S16 should be removed as it distracts from the much more important central message of the paper regarding the short peptides. Furthermore, the authors might find this an very interesting sample to investigate further in another study.

REVIEWERS' COMMENTS:

Note: All page and line references are to the manuscript with tracked changes visible.

Reviewer #1 (Remarks to the Author):

This work by Reynolds et al. is a thoroughly revised version of a manuscript that I have previously reviewed for Nature Nanotechnology. I have carefully read the rebuttal letter and the revised manuscript. The manuscript has been considerably strengthened by the additional experiments that were performed as well as the more quantitative analysis of the WAXS data that are now compared with simulations. Also, all the points that I had raised previously are addressed satisfactorily.

I would like to congratulate the authors for defining a new benchmark for the depth and quality of their study of short peptide self-assembly and for the combined use of cutting edge experimental and theoretical tools. I recommend publication of the manuscript in the current form.

We thank the reviewer for their very complimentary comments on this work.

Reviewer #2 (Remarks to the Author): Formally reviewer 3

In this revised manuscript, the authors have clarified important comments previously raised. They have also added stunning new data to further support their conclusions. As result, I think this manuscript is much improved. Following are few comments on the authors' reply to my previous comments.

1)

“This is an interesting point brought forward by the referee that has inspired us to undertake new work, which we believe, leads to a substantial improvement of the manuscript. To demonstrate that untwisted ribbons are amyloid crystals we now show cryoTEM images (new figure S7) from which electron diffraction occurs in proximity of the untwisted regions but not where the fibrils are twisted.

Furthermore, we have conducted also new, systematic experiments using Selected Area Diffraction (SAD) on flat ribbons (see figure 1 and S7 for representative examples): the flat ribbons lead in all cases to a single diffraction pattern, which as the referee will certainly appreciate, rules out any possible ambiguity concerning the crystallinity of these structures. This is further mirrored by a deeper analysis of WAXS spectra and high-resolution AFM images. Please refer to new figure 3 for a detailed characterization of these crystals. The conformation of crystallinity as proved by SAD observed electron diffraction is discussed on pg4, line 13.”

The authors added new SAD images that convincingly show crystal-like order in the proximity of the untwisted regions. They also write that they do not see any diffraction in the areas of the twisted fibrils, but I cannot find these images in the manuscript.

We apologise for the lack of clarity in the manuscript, here we were referring to the brightfield cryoTEM images which show enhanced contrast due to diffracting electrons from the crystals and semi-untwisted fibrils (Supplementary Figures 7a-d). We have changed the

wording slightly to emphasise this point (pg5, line 9) in addition we have added a new panel to Supplementary Figure 7 (d) clearly showing that whilst we can see diffraction contrast from the crystals it is absent in the helical ribbons.

The authors should add SAD images in the proximity to the twisted fibrils for comparison.

We have added two new panels showing the absence of diffraction patterns from the SAD experiments on fibrillar assemblies of ILQINS (Supplementary Figure 7i-j), and referred to them in the text pg 5, line 8.

Is it possible to calculate key molecular distances/lattice parameters using the diffraction patterns?

We did attempt to calculate lattice parameters from the SAD data and compare this to the WAXS data. However, this turned out to be challenging using the instrumentation available to us. The d-spacings provided by WAXS are averaged over all possible reflection angles, due to the incident X-ray beam hitting many randomly orientated peptide assemblies in an aqueous suspension. The diffraction patterns from the SAD images are collected from single crystals in a fixed orientation. In order to produce azimuthally averaged lattice parameters comparable with the WAXS data we would require a rotating stage on the TEM/SAD instrumentation.

2)

“We are actually in full agreement with the referee and we believe that this point arises from a misunderstanding, as we may have not sufficiently well explained this point in the original manuscript.

Molecular simulations give insight on the trajectories followed by the initial assembly of peptides to build up supramolecular species of given cross-sectional ratio (helical vs twisted ribbons). These simulations shed light on the temporal evolution of growth in the a, b and c axes, and on how the cross-section builds up until the elongation regime dominates the process (see figure 4a) up to around 10⁻² seconds. Thus, molecular dynamic simulations ONLY give the rationale to establish how the cross-sectional aspect ratio drives the fibrils into either twisted or helical ribbons. These MD simulations allow us to understand how pH, concentration and molecular mutations drive the assembly into either twisted ribbons or helical ribbons and simulations up to 10⁻² seconds are sufficiently long enough to establish this pathway.

The untwisting of twisted ribbons into crystals and the stability of helical ribbons are observed ONLY experimentally at much longer times, as the time scales involved with mesoscopic (and not molecular) rearrangements are much longer. Therefore simulations and microscopy experiments are fully complementary as they probe different time and length scales (MD: molecular length scale and sub-seconds time scales – AFM: mesoscopic length scale and minutes to hours/days time scales, see pg 17, line 1 for a discussion). Once crystals are formed from untwisted ribbons, lateral aggregation of crystallites may proceed without any significant energy barrier and we fully agree with the referee that this can undergo continuous lateral association at any time. One example is the aggregation captured in figure

5e. More examples have been added in the supplementary information (figure S15, discussed on pg 17, line 22). Conversely, lateral aggregation on twisted species, is much more expensive from an energetic point of view, since this implies additional torsional/bending energy. We therefore use MD simulations to unravel the build up of the aspect ratio in the early aggregates and once the aspect ratio trajectory is identified, we rely on AFM and TEM analysis to study the mesoscopic evolution. As the referee will note, we have, by means of the new AFM analysis, a resolution down to the single constitutive protofilament (figure 3j, S11, and S14). Thus, while the time scales between experiment and simulation cannot be matched, the length scales are now covered from the early molecular addition of the hexapeptides to the mesoscopic evolution of the fibrillar and crystalline species involved, validating the structural evolution followed by the individual aggregates across all the length scales involved in the process.”

The authors are probably correct in this specific case that there is likely a link between the observations made at the different timescales accessible to simulations and experiments, respectively. However, the authors’ evidence is nevertheless indirect, i.e. using information on points A and B to infer a pathway between A and B. Therefore, I suggest the authors to be much more careful in discussing the assembly mechanisms in the manuscript. For example on page 17: “The increased formation of anisotropic oligomers, at early time points (< 0.1 s), due to promotion of lateral aggregation in response to pH (pH 7 > pH 2) and chemical sequence (ILQINS>IFQINS>TFQINS) predicted by the simulations correlates extremely well with the experimental results collected at much longer timescales (hours/days).” Here “correlates extremely well with” should be changed to “is consistent with”.

We thank the reviewer for his/her comments and indeed have toned down some of the language relating to the assembly mechanisms. As suggested “correlates extremely well” altered to “is consistent with” (now on page 11, line 10). Pg 12, line 7 “nearly identical molecular structures” changed to “very similar molecular structures”. Pg12, line 14 “the mechanism of fibril-crystal interconversion” changed to “a mechanism of fibril-crystal interconversion”. Pg 12, line 19 inserted the word may into “amyloid crystal structures elucidated before MAY have relatable...”.

3)

“Again, we believe that this may be a misunderstanding. We indeed fully agree as the referee is inferring that the free energy barrier separating the two states (fibrils vs crystals) must be small, otherwise the transition would not be observed experimentally (although we have no elements to quantify how much the crystal minima is below the fibril one). We also fully agree that the exact depth of the free energy well of the dominant population must be dependent on the sequence and the conditions employed, and indeed the data presented in the manuscript precisely shows this. We do not understand, however, how the statement formulated above may be in contradiction with these findings: if we are able to capture the spontaneous untwist of fibrils into untwisted crystals without unfolding/refolding of the peptide, the energy of the final state (crystals) must be necessary lower than the initial state (fibril), since the only difference is the released torsional energy associated with the twist (which is positive by continuum mechanics). Therefore the lower minimum of the crystal compared to fibrils is not to be questioned and together with the previously supposed absolute minimum of amyloid fibrils in the protein folding energy landscape, this

immediately infers that a new absolute minimum has been found. This is in agreement with Referee 1 and 2 assessment and with earlier findings by Marshall et al. on GNNQQNY. The referee may also want to revisit figure 1 in Hartl et al. Nature Structure and Molecular Biology 2009 in terms of the favourable energetic transition fibrils → crystals.

We do agree with the referee that the original phrasing may benefit from clarification. To avoid ambiguity, we account for the remark of the referee and rephrase the text accordingly. We have added a conclusive paragraph in the manuscript (pg 19, line 7), stating: “From the experimental standpoint, the progression of the conversion from fibrils to more energetically stable crystals observed for ILQINS, IFQINS and TFQINS is consistent with the systematic switch of fibrils to crystals observed in GNNQQNY3, with the additional key finding that the two energetic states can be sampled by simply releasing torsional elastic energy but maintaining identical molecular packing, suggesting the energy barrier separating the two states to be very small.”

Here, the authors’ data show that the crystal-like structures are more favourable energetically than twisted filaments, but the difference in energy may be small. Thus, given all the conditions in this discussion that the authors added in their clarification, I think “introducing a new major paradigm” wording is far too exaggerated and should be removed. The authors should also tune down their wording and be more careful on discussing that the crystal-like structures may be energy minima etc.

Once again, we thank the reviewer for his/her comments and have made a further attempt to tone down the language when relating to the proposed newly discovered energy minima of the “amyloid crystal” in order to avoid any eventual unjustified claims. On Pg 12, line 22 we deleted “introducing a major new paradigm” as requested by the referee. on Pg12, line 25 we have changed the text from “This infers that amyloid crystals and not amyloid fibrils are the absolute minimum in the protein folding energy landscape.” to “This implies that amyloid crystals and not amyloid fibrils may occupy a newly discovered absolute minimum in the protein folding energy landscape” (“are” changed into “may occupy” and “the absolute minimum” into “a newly discovered absolute minimum”)

Reviewer #3 (Remarks to the Author): **Formally reviewer 2**

The manuscript by Reynolds et al describes the switch that can take place between fibrils and crystals. The paper is well written and I very much appreciate the careful consideration the authors have taken with the reviewer comments. The new section that analyses the WAX data is interesting and informative. I think it could have been even further improved by the added information afforded by fibre diffraction and this might have yielded insights into the differences between the structures.

We attempted to compare data from the SAD experiments with the WAXS reflections however due to the lack of a rotating stage on the TEM this turned out not to be conclusive. See above for a more detailed description.

The figure 3a shows comparisons between experimental and calculated data, which shows good fit, but not perfect. However, they are further supported by measurements from AFM. I wonder if the authors have considered the contribution of hydrogen bonding of the Q residues (which has been described before for polyQ peptides).

Contribution of hydrogen bonding of the Q residues is implicitly treated in the force field adopted in our molecular dynamics simulations. Q/N ladders are treated on the same footing as all other non-bonded interactions, via a combination of Coulomb, dispersion and solvation forces. They provide a strong but not irreplaceable contribution to axial stability via hydrogen bonding, and via sterics in the sheet-sheet interface (see Sawaya Eisenberg, Nature 2007 and Berryman..... Harris, Biophys J, 2011). Our work adds nothing to the by-now established corpus on amyloidogenicity or otherwise of specific residues, but discusses instead the kinetic effects of balancing the interaction strengths at different interfaces of the growing beta-sheet oligomers.

I am not sure where the 4.9 Å comes from? The hydrogen bonding distance for cross-beta structures is 4.76Å, but is often quoted as 4.7 or 4.8.

In relation to the hydrogen bonding distances from cross-beta structures we must respectfully disagree. This distance has been reported as 4.87 Å for GNNQQNY fibrils and crystals (see Nelson et al, Nature, 435, 773, 2005 and others), which is almost identical to the simulated distances calculated for ILQINS, IFQINS and TFQINS assemblies.

Finally, I appreciated all the changes to the paper, but I do not think that new Figure S16 is needed here. The authors claim (in their response letter) that this is the first crystal of 26 residues. However, whilst other crystal structures have not been solved (using Xray crystallography), several other amyloidogenic proteins/peptides form crystalline structures. PolyQ50 forms crystals amongst others. Figure S16 should be removed as it distracts from the much more important central message of the paper regarding the short peptides. Furthermore, the authors might find this an very interesting sample to investigate further in another study.

We first note that Figure S16 and the associated experimental effort, has been added in response to both referee 1 (i.e. in point 5 of the previous report he/she asked what happens for sequences longer than 6mers) and this same referee (formerly referee 2: from point 6 of his/her earlier report: “*And secondly, crystals can only be formed by a small subset of all peptides, namely the very short ones...*”). Thus, Figure S16 has been added as an improvement and we do not believe this distracts from the main message but it actually makes it only stronger. Furthermore, by removing it to please referee 3 we may very well upset referee 1.

With respect to polyQ structures, while these peptides do clearly form amyloid structures and many examples of polyQ containing crystals have been published, we can find no evidence in the literature of assembled “amyloid crystals” of polyQ. All of the published crystals seem to be either monomeric polyQ tracts bound to antibodies (Li, Nature Structural & Mol. Biol, 14, 381, 2007) or crystal structures of non-aggregated polyQ containing proteins such as Huntingtin (Kim et al, Structure, 17, 1205, 2009). If we have missed an important paper we apologise, but to the best of our knowledge the R3 fragment remains the largest amyloid crystal (possessing the defining intersheet and interstrand Bragg reflections), although no direct claim is made in the manuscript to this fact. Finally, polyQ peptides are hopolymers and as such they are very far from bearing general significance in the protein folding energy landscape.

Thus, unless the editor requires so, we propose to leave Supplementary Figure 16 in place as it provides strong evidence that the results of the study of the hexapeptides are not only of relevance to very short sequences, but bear a much broader significance.